

# Elucidating the boundary layer turbulence dissipation rate using high-resolution measurements from a radar wind profiler network over the Tibetan Plateau

Deli Meng[1], Jianping Guo[1,2]* Xiaoran Guo[1], Yinjun Wang[1], Ning Li[1], Yuping Sun[1], Zhen
Zhang[1], Na Tang[1], Haoran Li[1], Fan Zhang[1], Bing Tong[3], Hui Xu[1], Tianmeng Chen[1]
*[1]State Key Laboratory of Severe Weather, Chinese Academy of Meteorological Sciences,*
*Beijing 100081, China*
*[2]Fujian Key Laboratory of Severe Weather, Fujian Institute of Meteorological Sciences,*
*Fuzhou 350028, China*
*[3]State Key Laboratory of Urban and Regional Ecology, Research Center for Eco-*
*Environmental Sciences, Chinese Academy of Sciences, Beijing 100085, China*
*Correspondence to:* Dr/ Prof. Jianping Guo (Email: jpguocams@gmail.com)





## Abstract

The planetary boundary layer (PBL) over the Tibetan Plateau (TP) exerts a significant
influence on regional and global climate, while its vertical structures of turbulence and
evolution features remain poorly understood, largely due to the scarcity of observation.
This study examines the vertical profile and daytime variation of turbulence dissipation
rate ($\varepsilon$) in the PBL over the TP using the high-resolution (6 min and 120 m) measurements
from the radar wind profiler (RWP) network, combined with the hourly data from the
ERA5 reanalysis. Observational analyses show that the magnitude of $\varepsilon$ below 3km under
all-sky conditions exhibits large spatial discrepancy over the six RWP sites over the TP.
Particularly, the values of $\varepsilon$ at Minfeng and Jiuquan over the northern TP and Dingri over
the southern TP are roughly an order of magnitude greater than those at Lijiang, Ganzi and
Hongyuan over the eastern TP. This could be partially attributed to the difference of land
cover across the six RWP sites. In terms of the diurnal variation, $\varepsilon$ rapidly intensifies from
0900 local standard time (LST) to 1400 LST, and then gradually levels off in the late
afternoon. Under clear-sky conditions, both $\varepsilon$ and planetary boundary layer height ($z_i$) are
greater, compared with cloudy-sky conditions. This reveals that clouds would suppress the
turbulence development and deduce $z_i$. In the lower PBL ($0.2<z/z_i<0.5$, where $z$ is the
height above ground level), the dominant influential factor for the development of
turbulence is the surface-air temperature difference ($T_s - T_a$). By comparison, in the
upper PBL ($0.6<z/z_i<1.0$), both the $T_s - T_a$ and vertical wind shear (VWS) affect the
development of turbulence. Above the PBL ($1.0<z/z_i<2.0$), the shear production resulting
from VWS dominates the variation of turbulence. Under cloudy-sky conditions, clouds are
found to decrease the surface total solar radiation, thereby reducing $T_s - T_a$ and surface
sensible heat flux. This weakened sensible heat flux tends to inhibit the turbulent motion
within PBL especially in the lower PBL and decrease the growth rate of $z_i$. On the other
hand, the strong VWS induced by clouds enhances the turbulence above the PBL. The
findings obtained here underscore the importance of RWP network in revealing the fine-
scale structures of the PBL over the TP and gaining new insight into the PBL evolution.



## 1. Introduction


Turbulence ranks among the most intricate phenomena within the atmosphere, ensuring
that the planetary boundary layer (PBL) remains thoroughly mixed during daylight hours
(Li et al., 2023). As a result, the structure of the PBL is, to a considerable extent, governed
by the evolution of turbulence (Teixeira et al., 2021). Turbulence dissipation rate ($\varepsilon$)
reflects the amount of turbulent kinetic energy (TKE) that is converted into heat at the
Kolmogorov scale and is a measure of the turbulence intensity (McCaffrey et al., 2017;
Muñoz-Esparza et al., 2018). Proper parameterizations of the turbulence dissipation term
with the aid of observations have great impact on the model forecast skill for the weather
and climate, as $\varepsilon$ strongly affects vertical turbulent mixing through its influence on TKE
(Yang et al., 2017). Accurate estimation of $\varepsilon$ is crucial for understanding the structure of
turbulence in the PBL. To date, a variety of instruments have been used to observe or
retrieve the vertical profiles of $\varepsilon$, including sodar, radar wind profiler (RWP), radiosonde,
Doppler wind lidar (DWL) and ultrasonic anemometer (Dodson and Griswold, 2021;
Jacoby-Koaly et al., 2002; Kotthaus et al., 2023; Lv et al., 2021). Compared with the DWL,
the RWP exhibit better capability in capturing the turbulence structures in the cloudy sky.
Furthermore, it is hard for the radiosonde and ultrasonic anemometer to get the temporal
continuous measurements of the atmospheric turbulence, due to the high costs.
The Tibetan Plateau (TP), with an averaged elevation above 4000 m and an area of
approximately 2.5 million km$^2$, is towering into the lower and middle troposphere (Huang
et al., 2023). By receiving a greater amount of solar shortwave radiation, the surface layer
of the TP could transfer more heat through the PBL to the free atmosphere (Ma et al., 2023;
Wang et al., 2015). The PBL over the TP exhibits strong convective bubbling and upward
motions due to the lower air density and buoyancy effect, which results in significant
turbulence motions and turbulence-convection interactions with "popcorn" cloud structures
(Xu et al., 2002; Xu et al., 2023). Understand the statistical behavior of $\varepsilon$ is key to
revealing the vertical structure and evolution of PBL turbulence, which could improve the
parameterization of PBL processes over the TP (Ma et al., 2023; Wang et al., 2015; Xu et
al., 2019; Zhao et al., 2019). However, due to the limited observations of turbulence





profiles, the daytime variation characteristics of $\varepsilon$ over the TP and its main influential
mechanisms remains poorly understood.
A vast range of previous studies have attempted to figure out the mechanisms behind the
observed turbulence, but most of them are based on radiosonde measurements or model
simulation or reanalysis data (e.g., Banerjee et al., 2018; Che and Zhao, 2021; Wang et al.,
2023a). A myriad of driving mechanisms is proposed to account for the PBL development
over the TP, such as surface thermal and dynamic forcing, atmosphere stability, among
others (Chechin et al., 2023; Chen et al., 2016; Lai et al., 2021; Wang et al., 2023a; Wang
and Zhang, 2022). It has been demonstrated that the buoyancy term contribution on the
southern slope of the TP is significantly larger than that on the southeastern edge of the TP
(Wang et al., 2015). A larger surface-air temperature difference ($T_s - T_a$) and sensible heat
flux could promote the rapid growth of deep PBL in the western and southern TP (Chen et
al., 2013, 2016; Li et al., 2017a; Wang et al., 2016; Zhang et al., 2022). Chen et al (2016)
found that the weak atmosphere stability at the top of the mixed layer is a key factor
contributing to the rapid growth of the deep turbulence in winter over the TP.
Also, cloud radiative effects are found to be another significant factor to modulate the
evolution of daytime PBL turbulence (Bodenschatz et., 2010). The TP is characterized by
a high frequency of cumulus clouds which is about five times the regional mean over the
other areas of China (Wang et al., 2015), and the occurrence frequency of clouds over the
TP shows large diurnal and spatial variability, with the maxima in the afternoon in the
eastern TP (Wan et al., 2022). Guo et al. (2019) has revealed that the cloud tends to suppress
the development of summer PBL in the early afternoon across China using the fine-
resolution radiosonde observations. Under continuous cloudy-sky conditions, the
convective PBL develops slowly due to the smaller surface sensible heat compared to the
clear-sky conditions (Wang and Zhang, 2022). The turbulence motion and dynamic
structure in the PBL is contributed to the formation and development of the popcorn-like
convective clouds (Wang et al., 2020; Xu et al., 2002). Based on surface observation,
radiosonde, satellite and reanalysis data, Wang et al. (2020) pointed out that higher PBL
height and lower lifting condensation level due to lower temperature and lower atmospheric
density may enhance low cloud occurrence in the afternoon, and in turn influencing PBL
development over the TP. However, the turbulence structure differences between clear-sky



and cloudy-sky conditions are rarely explored, and the possible mechanism influencing the
cloud topped PBL turbulence evolution remains elusive. To the best of our knowledge,
most of the above-mentioned studies over the TP lack high-temporal resolution turbulence
profile observations. Coincidently, there exists a RWP network of China, which provides
us an invaluable opportunity to characterize the PBL turbulence structure over the TP (Guo
et al., 2021a). Therefore, the main objective of this study is to resolve the above issues over
the TP, by using observations from the RWP network together with other ground-based
meteorological measurements and the ERA5 data. We also analyze the joint effect of
thermodynamic and dynamic on $\varepsilon$ structure in the daytime (0900–1700 LST) PBL
through $T_s - T_a$ and vertical wind shear (VWS).

The remainder of this manuscript proceeds as follows, Section 2 describes the data and

methods used in this study. In Section 3, we analyze the spatio-temporal characteristics and
daytime pattern of $\varepsilon$ over the TP and investigate the possible thermodynamic and dynamic
effect on PBL turbulence under clear-sky and cloudy-sky conditions. The summary and
conclusions are given in section 4.

**2. Data and methods**
*2.1 The RWP network over the TP*

In this study, we use the vertical measurements of RWP data with a vertical resolution

of 120 m and a temporal resolution of 6-min from the RWP network over the TP, which
contains six operational sites (Minfeng, Jiuquan, Hongyuan, Ganzi, Lijang and Dingri)
operated by the China Meteorological Administration (CMA) during the period from
September 1, 2022 to November 31, 2023. The spatial distribution of RWP network over
the TP is shown in Fig. 1, and the detailed information for each RWP site, including
longitude, latitude, elevation, and land cover type is given in Table 1. Among these six
RWP sites, the Dingri site is located in the foothills of the Himalayas with an elevation
more than 4,300m above sea level (AGL), dominated by the land cover of bare and alpine
grassland. The Lijiang site is located in the southeastern TP characterized by complex



terrain with an elevation of about 2,400m AGL. The Ganzi and Hongyuan sites are situated
in the eastern TP, with elevations ranging from 3,300 to 3,500m AGL, and whose underlay
is mainly alpine grassland. The Minfen and Dunhuang sites are situated in arid and semi-
arid zones to the north of the TP, with elevations ranging from 1,400 to 1,500m, and their
dominant underlying land cover is mainly bare land. Therefore, these two sites are well
representative of the northern TP.
The RWP has the capability to obtain the high-temporal resolution atmospheric
turbulence and wind profiles over the TP compared to the radiosonde and reanalysis, which
makes it possible to analyze the fine PBL structures. The low and medium detection modes
of RWPs can acquire the wind field and turbulence information bellow 5 km AGL
(McCaffrey et al., 2017; Ruan et al., 2014). The RWP provides the radial observations
(marked as RAD subset), including profiles of radial velocity, doppler spectral width, and
signal noise ratio. Also provided by the RWP is the real-time sampling data (marked as
ROBS subset), including the profiles of horizontal wind (direction and speed), vertical
velocity, and refractive index structure constant (Liu et al., 2020). There exist large
uncertainties in the profiling measurements from RWP, thus the quality control for both
RAD and ROBS subsets are indispensable before retrieving related dynamic variables over
the TP (Liu et al., 2020; Wang et al., 2023). For instance, the profiling measurements highly
deviate from the truth below 0.5 km AGL and above 5 km AGL, which are attributed to
the near-surface clutter and significant beam attenuation, respectively (Guo et al., 2023).
Thus, here only the RWP measurements at heights from 0.5 km to 5 km are utilized for
analysis.

***2.2 Other miscellaneous meteorological data***
In this study, the hourly ground-based meteorological variables, including 2m air
temperature ($T_a$), ground surface temperature ($T_s$), pressure and cloud cover, are derived
from the six automatic weather sites over the TP. Also, 1-min rainfall observations from
rain gauges are used to minimize the potential influence of rainfall on the profiling
measurements from RWP. All these meteorological datasets are subjected to strict data-
quality control by the National Meteorological Information Center (NMIC) of the China




Meteorological Administration (Wang et al., 2023b). In addition, the hourly temperature
data at pressure levels from the ERA5 reanalysis data is also used in this study (Hersbach
et al., 2020).

***2.3 Methods***
*2.3.1 Retrieval of turbulence dissipation rate*
As a widely used ground-based equipment for detecting atmospheric wind profile (Liu
et al., 2020), RWP has the advantage to estimate $\varepsilon$ since it could measure Doppler velocity
spectrum in the radar volume where the turbulence parcel motion accounts for the spectral
width broadening (Jacoby-Koaly et al., 2002; White, 1999). In this study, the spectral width
method is applied to retrieve $\varepsilon$ from the RAD subset based on the hypothesis that
turbulence is isotropic, and the contributions to the spectral width from turbulent and non-
turbulent process are independent of each other (Solanki et al., 2021; White, 1999).
The major steps for $\varepsilon$ retrieval can be summarized as follows: (1) the spectral width
variance consisting of the turbulence and non-turbulence variance is obtained from the
spectral width measurements. (2) The non-turbulence broadening variances are
decomposed into beam broadening variance due to the finite width of the beam, shear
broadening variance generated by the presence of a wind gradient, and broadening variance
arising from data processing, among others (Nastrom, 1997). (3) The turbulent broadening
variance is extracted from the spectral width variance by excluding above mentioned non-
turbulence broadening variances. (4) $\varepsilon$ is estimated from turbulent broadening variance
with the main assumption of isotropic and homogeneous turbulence, as well as Gaussian
antenna symmetric illumination function and Gaussian radial response of the receiver
(White et al., 1999). For more details for the spectral width method, refer to the references
(Jacoby-Koaly et al., 2002; McCaffrey et al., 2017; Nastrom, 1997; Solanki et al., 2021).
*2.3.2 Estimation of planetary boundary layer height*
The planetary boundary layer height (hereafter referred to as $z_i$ ) is an important
parameter for characterizing fine vertical structure of the PBL, which has important
implications for the air mass exchange between Earth's surface and the atmosphere aloft,





thus affecting the cloud development and air pollutant dispersion (Dai et al., 2014; Dodson
and Griswold, 2021; Guo et al., 2021a; Li et al., 2017b; Wang et al., 2022).
Here daytime $z_i$ at each RWP site is derived from original signal-to-noise ratio (SNR)
profiles from the RAD subset based on the improved threshold method (ITM). The steps
are briefly outlined as follows. First of all, here we use the profile of normalized SNR
(NSNR) to avoid instrumental inconsistencies. Secondly, the NSNR threshold is set to 0.75
based on the $z_i$ estimated by the radiosonde measurements at the same site. Thirdly, the
profile of NSNR is scanned downwardly from the top to the ground surface. Finally, $z_i$
ultimately is determined as the height where the NSNR profile greater than 0.75 for the
first time. For more details for the ITM, refer to the references (Liu et al., 2019).
*2.3.3 Vertical wind shear*
The ROBS subset is used to calculate vertical wind shear (VWS), which is an important
parameter that present the dynamical effect on the development of PBL (Zhang et al., 2020).
VWS can be calculated as follows.

$$\text{VWS} = \left[\left(\frac{\partial u}{\partial z}\right)^2 + \left(\frac{\partial v}{\partial z}\right)^2\right]^{1/2} \tag{1}$$

where $u$ and $v$ denote zonal and meridional wind component, respectively, $z$ denotes the
sample height AGL.
*2.3.4 Classification of cloudy- and clear-sky conditions*
Using RWP combined with the ground-based cloud cover observations at each site, the
effect of clouds on daytime variations of PBL turbulence and $z_i$ over the TP are
investigated. Firstly, the 1-min precipitation and 6-min RWP data are time-matched to
remove the profile data half an hour before and after the precipitation to obtain non-
precipitation data (Wu et al, 2023). Then, all-sky conditions are defined as non-
precipitation hours. Finally, the clear-sky (cloudy-sky) conditions are identified as hours
with the cloud fraction less (greater) than 30% (80%), respectively (Guo et al., 2016;
Solanki et al., 2021).



*2.3.5 Calculation of the gradient Richardson number*
The evolution of turbulence in the PBL has been previously recognized to be closely
associated with atmospheric stability (Chechin et al., 2023; Chen et al, 2013; Lai et al.,
2021; Muhsin et al., 2016). Therefore, we take the gradient Richardson number ($Ri$) as a
variable to characterize atmospheric stability and the formation of turbulence over the TP.
Following Stull (1988), $Ri$ is formulated as follows:

$$Ri = \frac{g}{\theta_v} \frac{\partial \theta_v / \partial z}{(\partial u / \partial z)^2 + (\partial v / \partial z)^2} \tag{2}$$

where $\theta_v$ is the virtual potential temperature from ERA5, $u$ and $v$ are the hourly zonal
and meridional wind components derived from RWP, respectively, $g$ is the gravitational
acceleration, and $z$ represents the AGL.

## 3. Results and discussion

*3.1 Spatio-temporal distributions of daytime PBL turbulence dissipation rate*
Both the PBL turbulence dissipation rate and $z_i$ have significant diurnal variations over
mountain and urban areas (Adler et al., 2014; Liu et al., 2019; Solanki et al., 2021; Yang
et al., 2023). Since the longitude of the six sites over the TP is ranged from 82.7°E to
102.6°E, it is necessary to use the Local Standard Time (LST) to accurately capture the
daytime variations of the PBL and make a comparison between different sites.
Figure 2 presents a comprehensive overview of the $\varepsilon$ profile at 6 min intervals and
hourly averaged $z_i$ in lower troposphere at heights from 0.5 km to 3.0 km for six RWP
sites over the TP during the period from September 1, 2022 to November 31, 2023. The
magnitude of $\varepsilon$ and its vertical structures during the daytime at both Minfeng and Jiuquan
sites over the northern TP and at Dingri site over the southern TP stand in stark contrast to
those RWP sites (i.e., Lijiang, Ganzi and Hongyuan) in the eastern TP. As shown in the
right panels of Fig. 2, $\varepsilon$ generally decrease with increased height at all six RWP sites. It is
apparent that $\varepsilon$ exhibits a larger west-east and north-southern spatial difference under all-
sky conditions. In terms of the latitudinal variation, ε exhibits a decreasing trend from west



to east at both Minfeng and Jiuquan sites along the altitude belt of 38°N, so does the RWP
sites of Dingri, Lijiang, Ganzi and Hongyuan along the altitude belt of 30°N. In terms of
the meridional variation, $\varepsilon$ at the two RWP sites in the northern TP have a significantly
larger magnitude than the other four sites. In particularly, the maximum mean value of
daytime $\varepsilon$ in the height range of 0.5 to 3.0 km is found at Minfeng and Jiuquan in the
northern TP, which reaches up to $10^{-3.59}$ m$^2$ s$^{-3}$ and $10^{-3.73}$ m$^2$ s$^{-3}$, respectively. By
comparison, the least magnitude of $\varepsilon$ is found in the eastern TP, with the mean values of
$10^{-4.06}$ m$^2$ s$^{-3}$, $10^{-4.30}$ m$^2$ s$^{-3}$ and $10^{-4.22}$ m$^2$ s$^{-3}$ at Lijiang, Hongyuan and Ganzi, respectively.
Meanwhile, the mean magnitude of $\varepsilon$ at Dingri in the southern TP lies between the
magnitude of $\varepsilon$ in the northern and eastern TP, which is $10^{-3.88}$ m$^2$ s$^{-3}$.
The above results imply that the turbulence intensity at the RWP sites over the northern
and western TP is about one order of magnitude greater than that in the eastern TP. To
further investigate the possible reasons for this significant difference in $\varepsilon$, the relationships
between $T_s - T_a$ and $\varepsilon$ for different regions are presented in Fig. 3. The mean value of
$T_s - T_a$ in the northern and southern TP is 14.29°C, which is greater than that of eastern
TP with the value of 11.26°C (Fig. 3a). The mean daytime $\varepsilon$ for the two regions reaches
up to $10^{-3.74}$ m$^2$ s$^{-3}$ and $10^{-4.20}$ m$^2$ s$^{-3}$, respectively (Fig. 3b). Additionally, $\varepsilon$ is significantly
and positively correlated with $T_s - T_a$ (R>0.35, p<0.005), which illustrates that the
thermal forcing makes an important contribution to turbulence development in the TP (Figs.
3c and 3d).
Overall, the spatial distribution of the $z_i$ at all six RWP sites is clearly dependent on
geographical location (Fig. 2), which resembles that of the $\varepsilon$. The geographic pattern of
$z_i$ from RWP agrees well with those from radiosonde measurements (Che and Zhao, 2021)
and reanalysis (Slättberg, 2022). The land surfaces at both Minfeng and Jiuquan sites in
the northern TP are dominated by barren and relatively homogenous terrain, in sharp
contrast to the highly vegetated underlying terrain at both Ganzi and Hongyuan sites in the
eastern TP (Fig. 1). The sparse vegetation in the northern TP generally comes with large
Bowen ratio during the daytime, which tends to produce larger sensible heat flux compared
to that in the eastern TP. The increased turbulence intensity in PBL is generally associated
with larger sensible heat flux, which has been reported by previous studies (Wang et al.,





2016; Zhang et al., 2022). Thus, the spatial and magnitude of $\varepsilon$ over the TP are most likely
relevant to the underlying surface type.
Regarding the daytime pattern of turbulence (all six panels with color shading in Fig. 2),
the turbulence over the TP shows a pronounced signature of single–peak variability. During
the period 0900–1100 LST, the magnitude of $\varepsilon$ at all six RWP sites is relatively weak.
From 1100 LST onward, with the increase of downward solar shortwave radiation, surface
sensible heat flux gradually rises, which leads to acceleration of turbulence mixing
processes. Then, $\varepsilon$ reaches peak in the early afternoon (1300–1500 LST). Afterwards,
during the later afternoon (1500–1700 LST), $\varepsilon$ diminishes gradually. Likewise, $z_i$ almost
follows the same daytime variation pattern of $\varepsilon$.

*3.2 Characteristics of daytime PBL turbulence dissipation rate under clear- and*
*cloudy-sky conditions*
The influence of clouds on PBL has been discussed and analyzed in previous studies
(e.g., Guo et al., 2016; Huang et al., 2023; Ma et al., 2023; Schumann et al., 1991; Yu et
al., 2004). To reveal the potential impact of clouds on PBL $\varepsilon$ over the TP, the comparison
analyses between clear- and cloudy-sky conditions are presented in this section. Figure 4
shows the daytime cycle of mean $\varepsilon$ profile and $z_i$ averaged over the six RWP sites under
all-, clear- and cloudy-sky conditions. Overall, both the profile of $\varepsilon$ and $z_i$ over the TP
present distinct single–peak variations, and their peaks approximately occur at 1400 LST
(Fig. 4a). The daytime averaged $\varepsilon$ below 3.0 km AGL is $10^{-3.95}$ m$^2$ s$^{-3}$, and mean $z_i$ is
1472 m, respectively. There is a significant positive correlation between $\varepsilon$ and $z_i$ during
the daytime (R=0.63, p<0.01).
Under clear-sky condition, the daytime mean $\varepsilon$ is $10^{-3.88}$ m$^2$ s$^{-3}$ (Fig. 4b). During the
period 1300–1500 LST, $\varepsilon$ is ranged from $10^{-3.43}$ to $10^{-2.82}$ m$^2$ s$^{-3}$ ($10^{-4.17}$ to $10^{-3.40}$ m$^2$ s$^{-3}$) at
heights from 0.5 km (1.0 km) to 1.0 km (2.0 km) in lower (upper) PBL. Thus, the well-
mixed turbulence maintains the development of PBL in the early afternoon. Under cloudy-
sky condition (Fig. 4c), the daytime mean value of $z_i$ can reach up to 1415 m, which is
117 m lower than that of clear-sky conditions. This means that the clouds would suppress





the development of the PBL turbulence in the early afternoon which has been observed by
the radiosonde observations described in Guo et al (2016).

Furthermore, the probability density distribution (PDF) of $\varepsilon$ in PBL ($0.2<z/z_i<1.0$) and

above the PBL ($1.0<z/z_i<2.0$) under all-, clear- and cloudy-conditions are given in Fig. 5.
Here AGL ($z$) is normalized by $z_i$ to provide a nondimensional vertical coordinate of $z/z_i$.
Overall, the mean $\varepsilon$ are $10^{-3.82}$, $10^{-3.79}$ and $10^{-3.85}$ m$^2$ s$^{-3}$ at height of $0.2<z/z_i<2.0$ under all-,
clear- and cloudy-sky conditions, respectively (Fig. 5a). Within the PBL (Fig. 5b), the mean
$\varepsilon$ under clear-sky conditions ($10^{-3.27}$ m$^2$ s$^{-3}$) is greater than that of under cloudy-sky
conditions ($10^{-3.36}$ m$^2$ s$^{-3}$), and the standard deviation of $\varepsilon$ under clear-sky conditions is
slightly greater than that under cloudy-sky conditions. This illustrates that clouds can
significantly inhibit the turbulence intensity in the PBL, with the value of $\Delta\varepsilon$ between
clear- and cloudy-sky conditions is $-10^{-4.0}$ m$^2$ s$^{-3}$. However, above the PBL (Fig. 5c), $\varepsilon$
presents normal distribution characteristics, and there is no significant difference between
the mean $\varepsilon$ under clear- and cloudy-sky conditions.

To examine the impact of clouds on the vertical structure of turbulence within and above

the PBL, Figure 5d shows the normalized contoured frequency by altitude diagram
(NCFAD) of the $\Delta\varepsilon$ for normalized ($z/z_i$) profiles of $\varepsilon$ between cloudy-sky and clear-sky
conditions. Within the PBL, $\Delta\varepsilon$ is negative, and $|\Delta\varepsilon|$ generally decrease with increased
$z/z_i$, where $\Delta\varepsilon$ is $-10^{-4.3}$ m$^2$ s$^{-3}$ at $z/z_i=0.5$, and $-10^{-5.0}$ m$^2$ s$^{-3}$ at $z/z_i=1.0$, respectively.
This suggests that clouds may weaken turbulence within the PBL (Figs. 4b and 4c),
especially in the lower PBL ($z$=820m, $z/z_i<0.5$).

### *3.3. Potential factors Influencing daytime PBL turbulence dissipation rate*

#### *3.3.1 Surface-air temperature difference*

The vertical structure of PBL $\varepsilon$ and $z_i$ over the TP show obvious spatial differences in

the context of a complex subsurface. The diverse land cover types lead to differences in
surface albedo and soil moisture, which in turn lead to distinctions in thermodynamic
characteristics such as sensible heat flux (Ma et al., 2023). Buoyant production driven by
solar heating from the surface is one of the dominant sources generating turbulence in the





convective PBL. The surface sensible heat flux is an important thermodynamic factor that
affects the buoyant convective processes (Stull, 1988). Meanwhile, previous studies (e.g.,
Wang et al., 2022; Yang et al., 2023) have suggested that $T_s - T_a$ can serve as a good
proxy for the sensible heat flux. There are not sensible heat flux measurements at six RWP
sites in this study, and thus we directly take $T_s - T_a$ as a proxy thermodynamic variable to
analyze its potential connection to variation of PBL turbulence.
Figure 6 shows the magnitude of $\varepsilon$ varies as a function of $T_s - T_a$ for all six sites,
within (0.2<$z/z_i$<1.0) and above (1.0<$z/z_i$<2.0) the PBL, under all, clear- and cloudy-sky
conditions, respectively. $T_s - T_a$ are first classified into five bins, which are then
statistically analyzed against the corresponding $\varepsilon$ averaged for $z/z_i$ values between 0.2
and 2.0 to obtain regression equations incorporating slopes. Further, Table 2 shows the
scatter plots between $Log_{10}\varepsilon$ and $T_s - T_a$ (and VWS) at different altitude ranges under
all-, clear- and cloudy-sky conditions. $Log_{10}\varepsilon$ is found to be linearly correlated with $T_s -$
$T_a$ (and VWS) (p<0.05). The surface sensible heat flux generally increases with increased
$T_s - T_a$, thus the increased $T_s - T_a$ intensifies the turbulence in PBL (0<$z/z_i$<1.0), which
is shown in Fig. 6b, e, h. Within the PBL, $\varepsilon$ is also positively correlated with $T_s - T_a$
whose slope values are larger than those at 0.2<$z/z_i$<2.0. As $T_s - T_a$ rises, the larger
surface sensible heat flux would lead to enhanced buoyancy process and turbulent motion
within the PBL. On the other hand, $\varepsilon$ above the PBL is negatively correlated with $T_s -$
$T_a$ (Figs. 6c, f, i). This suggests that $T_s - T_a$ dramatically affects the development of
turbulence within the PBL, whereas it has little effect on the turbulence above the PBL.
Within the PBL, the magnitude of slope (slope=0.019) under clear-sky conditions is
larger than that of under-cloudy conditions (slope = 0.015) as shown in Figs. 6e and 6h.
This implies that $T_s - T_a$ is one of the dominant factors affecting the PBL turbulence,
particularly under the clear-sky conditions. Given that turbulence in the mixed PBL over
the TP is usually driven by convection (Xu et al., 2023), as $T_s - T_a$ decreases when clouds
are present, less heat is transferred from the surface to the atmosphere, reducing the
buoyancy flux and leading to weaker turbulence in the PBL, especially for the lower PBL
(0.2<$z/z_i$<0.5), as shown in Figures 4b and 4c. Consequently, the clouds tend to suppress
the development of PBL (Fig. 5a) and reduce $z_i$.

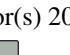



*3.2.2 Vertical wind shear*
Besides $T_s - T_a$, VWS is another crucial dynamic parameter that is related to the
mechanical turbulence within the PBL. Similar to Fig. 6, Figure 7 presents the relationship
between $\varepsilon$ and VWS (both normalized by $z_i$) within and above the PBL under all-, clear-
and cloudy-sky conditions, respectively. The near-surface clutter significantly increases
the uncertainty of RWP data, which leads to incapability of analyzing the effect of wind
shear on $\varepsilon$ below 0.5 km AGL in the following sections.
Regardless of within or above the PBL, $\varepsilon$ is positively correlated with VWS as shown
in Fig. 7a, d, g and Tabel 2, which indicates that larger VWS leads to stronger turbulence.
This suggests that the dynamic effect of VWS promotes the development of turbulence.
Within the PBL (Figs. 7b, e, h), the slope of $\varepsilon$ against VWS are smaller than at
$0.2<z/z_i<2.0$ with values ranging from 9.5 to 10.3. Above the PBL (Figs. 7c, f, i), the
values of the slope are larger with values ranging from 10.7 to 18.1, which demonstrating
that the dynamical effects of VWS influence the development of turbulence both within
and above the PBL.
Under cloudy-sky conditions (Figs. 7h, i), the effect of VWS on turbulence within the
PBL (Slope =10.3) is weaker than above the PBL (Slope=18.1), significantly. Compared
to the clear-sky conditions (Figs. 7e, f), the values of the slopes are larger for that of under
cloudy-sky conditions (Figs. 7h, i) both within and above the PBL. Remarkably, above the
PBL, the effect of clouds on turbulence is more dramatic, as the slope value under cloudy-
sky conditions is nearly twice as large as that of under clear-sky conditions. These results
indicate the significant mechanical processes driven by VWS is important in the
development of turbulence. A larger VWS in the PBL corresponds to stronger turbulence.
Besides, above the PBL, the mechanical process of VWS is enhanced under cloudy-sky
conditions.

*3.2.3. Joint influence of $T_s - T_a$, VWS and atmosphere stability on $\varepsilon$*
It was stated that turbulence can be produced by buoyant convective processes (i.e.,
thermals of warm air rising) and by mechanical processes (i.e., wind shear). From the



previous section, it is known that $T_s - T_a$ and VWS both affect the development of PBL
turbulence. Figure 8 gives the slope profiles of $\varepsilon$ against $T_s - T_a$ and VWS at normalized
heights ($z/z_i$) under all-, clear- and cloudy-sky conditions, respectively.
As inferred from the previous findings, $T_s - T_a$ primarily influences turbulence
development within the PBL, irrespective of clear-sky and cloudy-sky conditions (Fig. 6).
Figure 8a shows that the slope values within the PBL are predominantly positive, and the
slope value decreases rapidly with height, which indicates that the influence of $T_s - T_a$ on
PBL turbulence experiences decreasing trend with height. Interestingly, there is a nearly
linear of slope value from the lower PBL to a smaller positive value near the top of the
PBL. Above the PBL, the slope value becomes negative. This may be due to the linear
decrease of heat flux transport and buoyancy term in the convective PBL (Stull, 1988).
Therefore, these findings highlight the predominant thermal forcing of $T_s - T_a$ on
turbulence development within the lower PBL. In addition, when 0.2<$z/z_i$<0.5, the slope
values are larger for clear-sky conditions than for cloudy-sky conditions, while there is
little difference for the clear-sky and cloudy-sky conditions when 0.5<$z/z_i$. Hence, under
clear-sky conditions, the thermodynamic effect of $T_s - T_a$ is more pronounced within the
lower PBL.
As shown in Fig. 7, it is evident that VWS influences turbulence development within
and above the PBL. Figure 8b shows that when 0.2<$z/z_i$<2.0, the slope values are
consistently positive, indicating that VWS predominantly affects turbulence development
within the mid-, upper- PBL and above the PBL. Moreover, when 0.2<$z/z_i$<1.2, the slope
values increase with height. However, when 1.4<$z/z_i$<2.0, the slope values exhibit a
decreasing trend, which suggesting a diminishing influence of VWS. Additionally, within
the PBL (0.2<$z/z_i$<0.7), the slope values under clear-sky conditions are close to those
under cloudy-sky conditions, while the slope values under cloudy-sky conditions are even
greater when 0.7<$z/z_i$<2.0. For instance, when $z/z_i$=1.4, Slope$_{Clear-sky}$=14.6, while
Slope$_{Cloudy-sky}$=27.0, indicating that the latter is 1.8 times larger than the former. These
results suggest that clouds are primarily responsible for enhancing mechanical processes
from VWS on turbulence within the upper PBL and above the PBL.



Furthermore, it can be concluded that, $T_s - T_a$ is the thermodynamic factor influencing
turbulence development within the lower PBL ($0.2<z/z_i<0.5$), both $T_s - T_a$ and VWS
jointly strengthen turbulence development in the upper PBL ($0.6<z/z_i<1.0$), and VWS
emerges as the predominant factor affecting turbulence development above the PBL
($1.0<z/z_i<2.0$) (Figs. 8a, b).
The previous sections have revealed that hours of both high $T_s - T_a$ and strong wind
shear would strengthen the turbulence within the PBL. Therefore, it's necessary to analyze
the combined influence of thermodynamics and dynamics factors on the development of
turbulence. Figure 9 presents the joint distribution of $\varepsilon$ with $T_s - T_a$ and VWS within and
above the PBL under all-, clear- and cloudy-sky conditions. Within the PBL (Figs. 9b, e,
h), higher $T_s - T_a$ and VWS correspond to stronger turbulence (Fig. 8). In contrast, the
thermodynamic effect of $T_s - T_a$ on turbulence has diminished and is no longer a
dominant factor above the PBL, while the dynamical effect of VWS becomes the dominant
factor (Figs. 9c, f, i). Compared to clear-sky conditions, both $T_s - T_a$ and VWS decrease
under cloudy-sky conditions (Fig. 9h). This means that the weakening of both
thermodynamic and dynamic effects leads to a decrease in turbulence, thereby inhibiting
the development of turbulence within the PBL. Therefore, under cloudy-sky conditions,
although the VWS is reduced, the dynamical effect of VWS on turbulence is strengthened
(Figs. 7i and 8b), which in turn strengthens turbulence.
Since buoyant and mechanistic forcing jointly influence the turbulence within the PBL,
and VWS only represents the dynamic driving effect, it cannot accurately portray the effect
of thermodynamic and dynamic effects on the PBL turbulence. The gradient Richardson
number ($Ri$), on the other hand, is one of the important parameters characterizing
atmospheric stability and can compare the buoyant turbulence production term and the
shear production term in the form of a dimensionless ratio.
Similar to Fig. 9, the joint distribution of $\varepsilon$ with $T_s - T_a$ and $Ri$ within and above the
PBL under all-sky, clear-sky and cloudy-sky conditions is given in Fig. 10. Within the PBL
(Figs. 10b, e, h), it is evident that the turbulence in the PBL is enhancing for unstable
conditions. Furthermore, under clear-sky conditions (Fig. 10e), the maximum number of
samples is found when $Ri<1.0$ and $T_s - T_a>21.1$ in strongly unstable conditions caused





by the buoyancy forcing driven by the larger $T_s - T_a$. While the effect of $Ri$ on turbulence
is relatively weakened above the PBL (Figs. 10c, f, i).

**4 Summary and concluding remarks**

This study investigates the characteristics of spatio-temporal distribution of daytime
PBL turbulence dissipation rate ($\varepsilon$) based on more than one-year record (September 2022–
October 2023) of profiling measurements from a radar wind profilers (RWP) network on
the Tibet Plateau (TP). Also analyzed are the evolution of $\varepsilon$ in the PBL and the possible
influential mechanisms.
First of all, $\varepsilon$ is firstly retrieved from the vertical wind measurements from RWP using
the spectral width method. Afterwards, the spatial pattern of $\varepsilon$ is examined. Results shows
that the values of $\varepsilon$ at both Minfeng and Jiuquan sites in the northern TP, and at Dingri
over the southern TP are about one order of magnitude greater than those at the RWP sites
of Lijiang, Ganzi and Hongyuan over the eastern TP. Coincidently, Minfeng and Junquan
are dominated by bare or semiarid land, as opposed to the highly vegetation-covered land
surface at Lijiang, Ganzi and Hongyuan. This suggests the spatial discrepancy of $\varepsilon$ over
the TP is highly relevant to the types of underlying land cover.
Although $\varepsilon$ exhibits a variety of magnitudes among the six RWPs, the daytime pattern
and vertical structure of $\varepsilon$ are similar. Turbulence reaches the peak in the early afternoon
(1300–1500 LST), coinciding with the highest PBL top. Under cloudy-sky conditions, the
daytime mean value of $\varepsilon$ is $10^{-4.02}$ m$^2$ s$^{-3}$, and the daytime mean value of the PBL height
($z_i$) can reach up to 1415 m, which is 117 m lower than that of clear-sky conditions,
indicating that clouds would suppress the development of the PBL turbulence.
As far as both the thermodynamic and dynamic forcings are concerned, surface-air
temperature difference ($T_s - T_a$) and vertical wind shear (VWS) variables are examined by
performing correlation analysis with $\varepsilon$. The slope values of $\varepsilon$ against $T_s - T_a$ under clear-
sky conditions is larger (slope=0.019) than under-cloudy conditions (slope = 0.013) within
the PBL, while those values are negative above the PBL. The slope values of $\varepsilon$ against
VWS is positive regardless of within or above the PBL, where the largest value of 18.1 is





observed above the PBL under cloudy-sky conditions, and the smallest value of 9.5 is
observed in the PBL within clear-sky conditions.
Both the thermodynamic effect of $T_s - T_a$ and the dynamic effect of VWS enhance the
development of turbulence under clear-sky or cloudy-sky conditions in the PBL. In the
lower PBL ($0.2 < z/z_i < 0.5$), $T_s - T_a$ has a larger positive slope with $\varepsilon$, which suggests that
thermal forcing emerges as the dominant factor influencing development of the turbulence
and PBL. By comparison, in the upper PBL ($0.6 < z/z_i < 1.0$), $T_s - T_a$ and VWS jointly
influence the development of turbulence, with larger $T_s - T_a$ leading to unstable
atmospheric stability and stronger turbulence. Above the PBL ($1.0 < z/z_i < 2.0$), VWS
becomes the dominant factor influencing the development of turbulence. Compared to
clear-sky conditions, on one hand, clouds would diminish $T_s - T_a$, resulting in decreased
heat transfer from the surface to the PBL top, thereby weakening turbulence within the
lower PBL ($0.2 < z/z_i < 0.5$), inhibiting PBL development, and decreasing $z_i$. On the other
hand, the stronger wind shear process would enhance the turbulence above PBL under the
cloudy-sky conditions.
Although the above-mentioned findings of the PBL turbulence over the TP are the first
results from profiling network observations to the best of our knowledge, fine-resolution
spatial distribution remains unclear, largely due to the sparse distribution of RWP network
on the TP. On top of this, the role of roughness length remains known in the variation and
evolution of turbulence, especially in the lowest part of PBL, which warrants a field
campaign involved in the high-density turbulence observation network along with high-
resolution satellite images.

**Data Availability**
The authors would like to acknowledge the National Meteorological Information Centre
(NMIC) of China Meteorological Administration (CMA) (https://data.cma.cn) for
providing the high-resolution radar wind profiler and ground-based meteorological data,



which can be only accessed via registration. We are grateful to ECMWF for providing
ERA5 hourly data (https://www.ecmwf.int/en/forecasts/datasets/reanalysis-datasets/era5/).

## Acknowledgments

This work was jointly supported by the National Natural Science Foundation of China
under grants 42325501, U2142209 and 42105090. Last but not least, we appreciated
tremendously the constructive comments and suggestions made by the anonymous
reviewers that significantly improved the quality of our manuscript.

## Author Contributions

The study was completed with close cooperation between all authors. JG conceived of the
idea for this work. DM performed the analysis, DM and JG drafted the original manuscript
with contributions from XG, NL, YS, ZZ and NT. YW, HL, FZ, BT, HX and TC provided
useful suggestions and comments for the study and helped revise the manuscript.

## Completing interests

The authors declare that they have no conflict of interest.

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



**Table list**
**Table 1.** Summary of the geographical conditions and land surface of the six radar wind
profiler (RWP) sites over the Tibet Plateau (TP).

| RWP site | Latitude (°E) | Longitude (°N) | Elevation (m) | Land cover types |
|---|---|---|---|---|
| Minfeng | 82.69 | 37.07 | 1408.9 | Bare land |
| Jiuquan | 98.49 | 39.77 | 1477.2 | Bare land |
| Dingri | 87.07 | 28.39 | 4326.0 | Grassland |
| Ganzi | 100.00 | 31.62 | 3353.0 | Bare land, grassland |
| Hongyuan | 102.55 | 32.79 | 3465.0 | Bare land, grassland |
| Lijiang | 100.22 | 26.85 | 2382.4 | Bare land, grass land |




**Table 2.** Summary of the correlation of $Log_{10}\varepsilon$ at different altitude ranges under all-,
clear- and cloudy-sky conditions with $T_s - T_a$ and vertical wind shear (VWS) for all six
RWP sites. The superscript * for R indicates that the regression slope is statistically
significant at $p < 0.01$.

| Conditions | $Log_{10}\varepsilon$ VS $T_s - T_a$ | $Log_{10}\varepsilon$ VS VWS |
|---|---|---|
| all-sky, $0.2<z/z_i<2.0$ | y=0.010x–4.05, R= 0.21* | y=13.6x–4.19, R=0.29* |
| all-sky, $0.2<z/z_i<1.0$ | y=0.018x–3.70, R=0.29* | y=13.2x–3.77, R=0.20* |
| all-sky, $1.0<z/z_i<2.0$ | y=-0.005x–4.20, R=-0.09* | y=17.6x–4.57, R=0.36* |
| clear-sky, $0.2<z/z_i<2.0$ | y=0.010x–4.03, R=0.23* | y=10.7x–4.13, R=0.26* |
| clear -sky, $0.2<z/z_i<1.0$ | y=0.018x–3.67, R=0.30* | y=11.1x–3.70, R=0.17* |
| clear -sky, $1.0<z/z_i<2.0$ | y=-0.006x–4.16, R=-0.12* | y=13.8x–4.52, R=0.34* |
| cloudy-sky, $0.2<z/z_i<2.0$ | y=0.009x–4.06, R=0.17* | y=18.5x–4.29, R=0.33* |
| cloudy-sky, $0.2<z/z_i<1.0$ | y=0.018x–3.72, R=0.26* | y=15.5x–3.84, R=0.23* |
| cloudy-sky, $1.0<z/z_i<2.0$ | y=-0.004x–4.23, R=-0.08* | y=26.2x–4.67, R=0.42* |




**Figures**

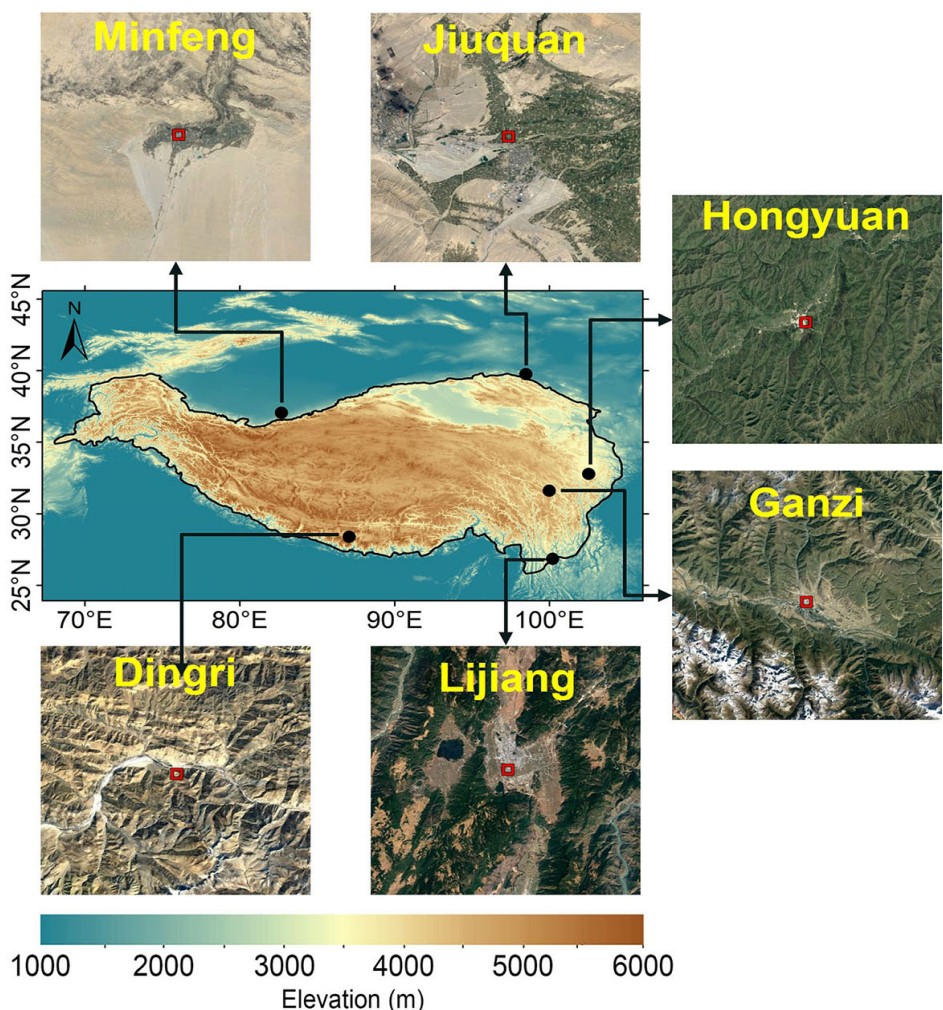

**Figure 1.** Spatial distribution of radar wind profiler (RWP) network comprised of six sites (in black solid circles) on the Tibetan Plateau (TP). The inset map surrounding the main frame denotes the RGB satellite image from © Google Earth that is centered at each RWP site.

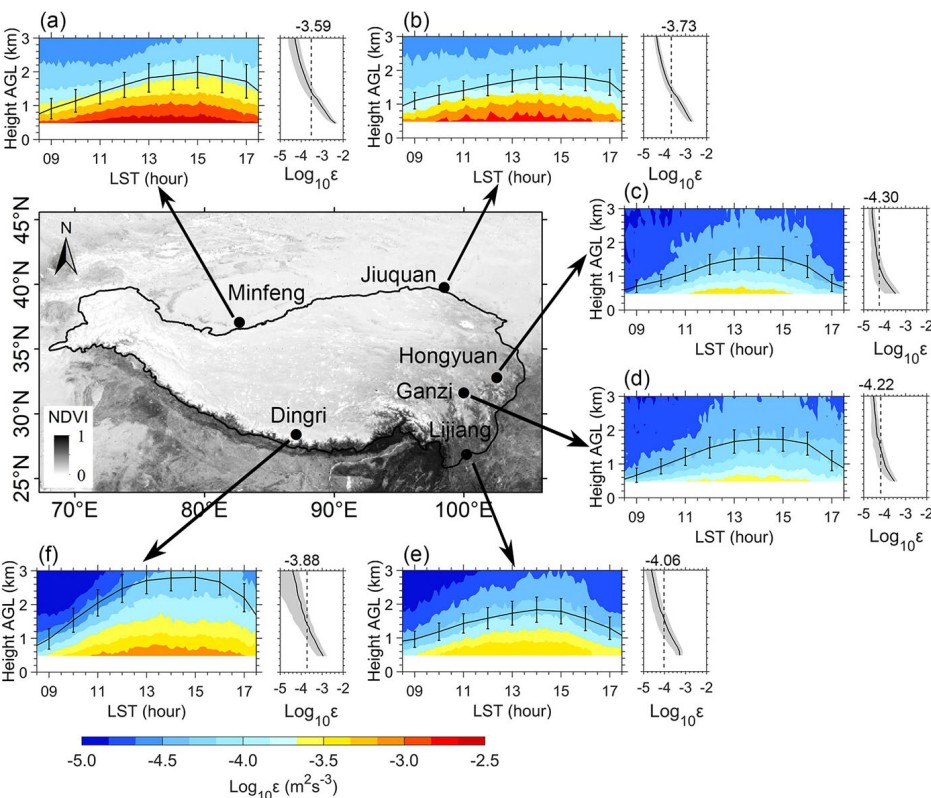

772

**Figure 2.** Spatial distribution of the diurnal evolution of the vertical profile of logarithmic turbulence dissipation rate ($Log_{10}\varepsilon$ in color shading, unit: m$^2$ s$^{-3}$) at 120 m vertical resolution and 6 min intervals, and hourly mean planetary boundary layer height ($z_i$, black line, unit: km) during daytime under all-sky conditions from 0900 to 1700 LST for the period September 2022 to October 2023 as retrieved from the profiling measurements at six RWP sites over the TP. The vertical bars indicate the 0.5 standard deviations for $z_i$. Also shown on the right-hand side panel are temporally averaged vertical profile of $\varepsilon$ (black line) and its corresponding one standard deviation (gray shading).

781

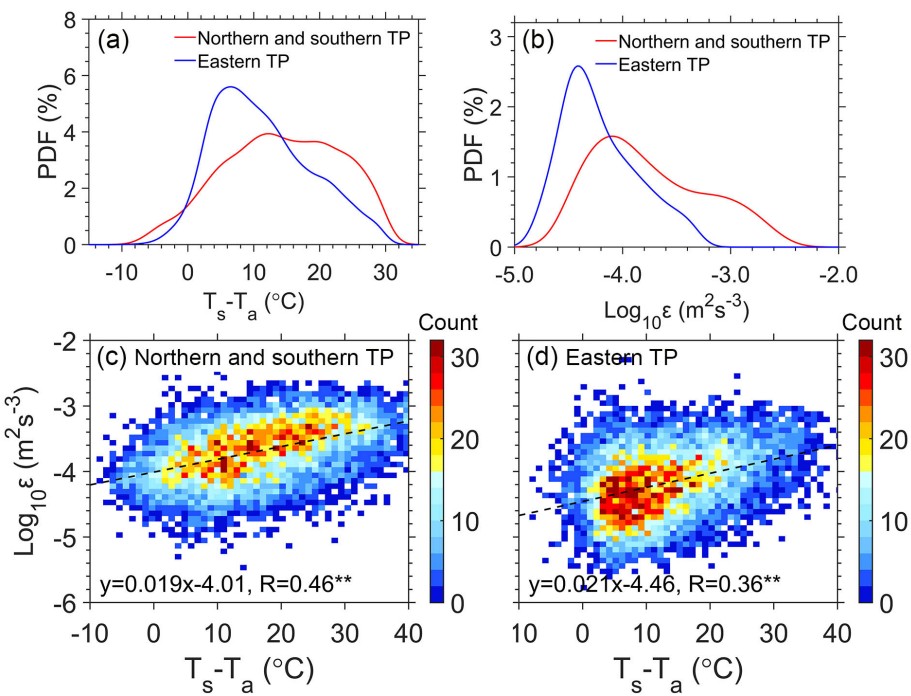

**Figure 3.** (a) PDF of surface-air temperature difference $(T_s - T_a)$ for the northern and southern TP (red line) and eastern TP (blue line), (b) same as (a), but for PDF of $Log_{10}\varepsilon$ estimated from the measurements of radar wind profilers (RWPs) at the height below 0.5 km and above 5 km, (c) scatter plots of $Log_{10}\varepsilon$ as a function of $T_s - T_a$ in the northern and southern TP, (d) same as (c), but for the eastern TP during daytime under all-sky conditions from 0900 to 1700 local standard time (LST) for the period September 2022 to October 2023. The superscript ** for R indicates that the regression slope is statistically significant at $p < 0.01$ level.

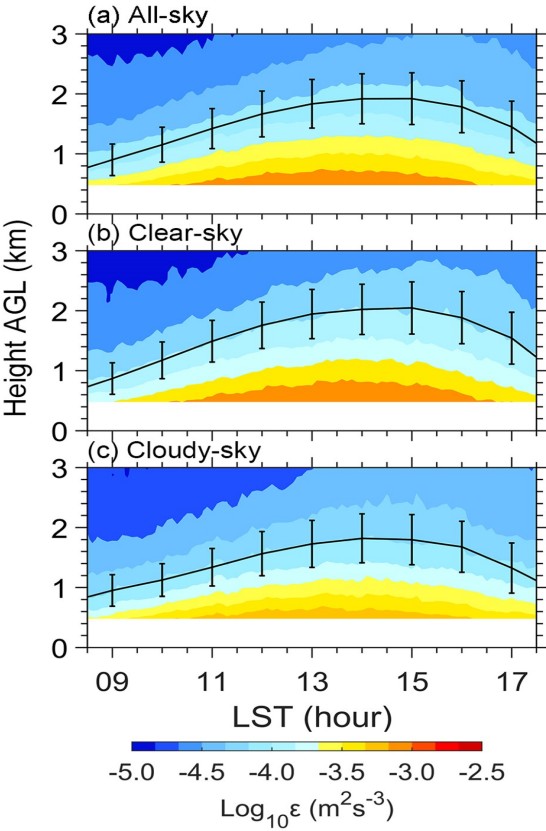

**Figure 4.** Diurnal evolution of the vertical profile of $Log_{10}\varepsilon$ (color shading, unit: m$^2$ s$^{-3}$) and $z_i$ (solid line, unit: km) averaged over the six RWP sites over the TP during daytime from 0900 to 1700 LST for the period September 2022 to October 2023 for (a) all-sky conditions, (b) clear-sky conditions and (c) cloudy-sky conditions. The vertical bars indicate the 0.5 standard deviations.

800

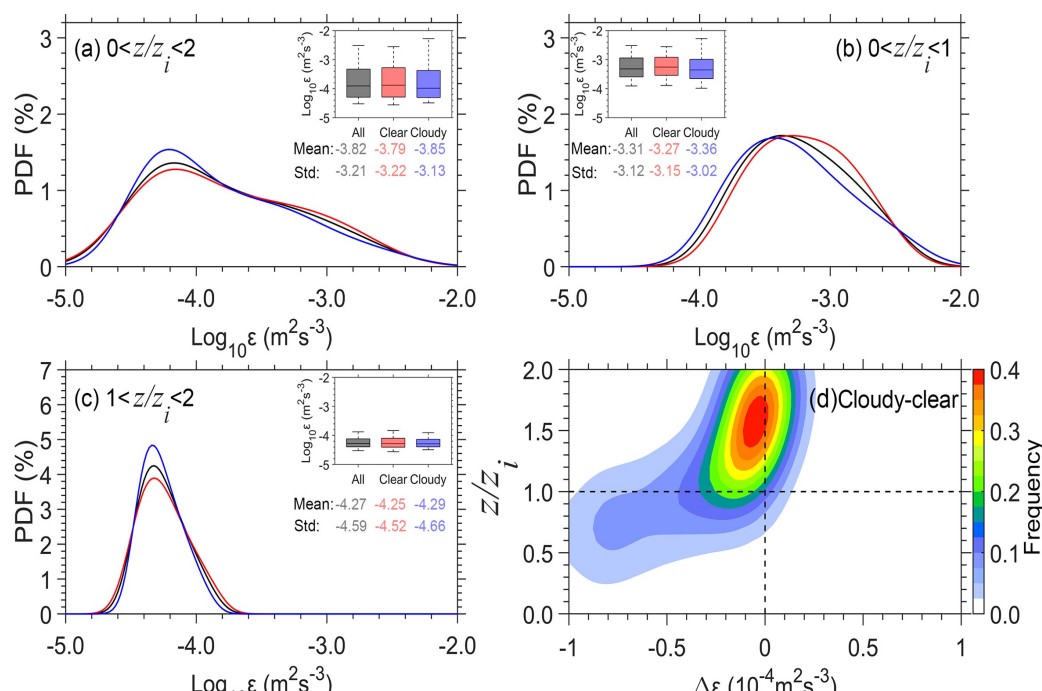

801

**Figure 5.** PDF of daytime $Log_{10}\varepsilon$ (a) in the whole lower troposphere ($0.2 < z/z_i < 2.0$), (b) in the PBL ($0.2 < z/z_i < 1.0$) and (c) above the PBL ($1.0 < z/z_i < 2.0$) over the TP under all-sky (black), clear-sky (red) and cloudy-sky (blue) conditions, respectively. (d) Normalized contoured frequency by altitude diagram (NCFAD) for the difference of $\varepsilon$ between cloudy-sky and clear-sky conditions ($\Delta\varepsilon$) over the TP. Note that $z_i$ denotes the depth of the PBL, the height ($z$) and turbulence dissipation rate ($\varepsilon$) is normalized by $z_i$ in order to give a nondimensional vertical coordinate in the form of $z/z_i$.



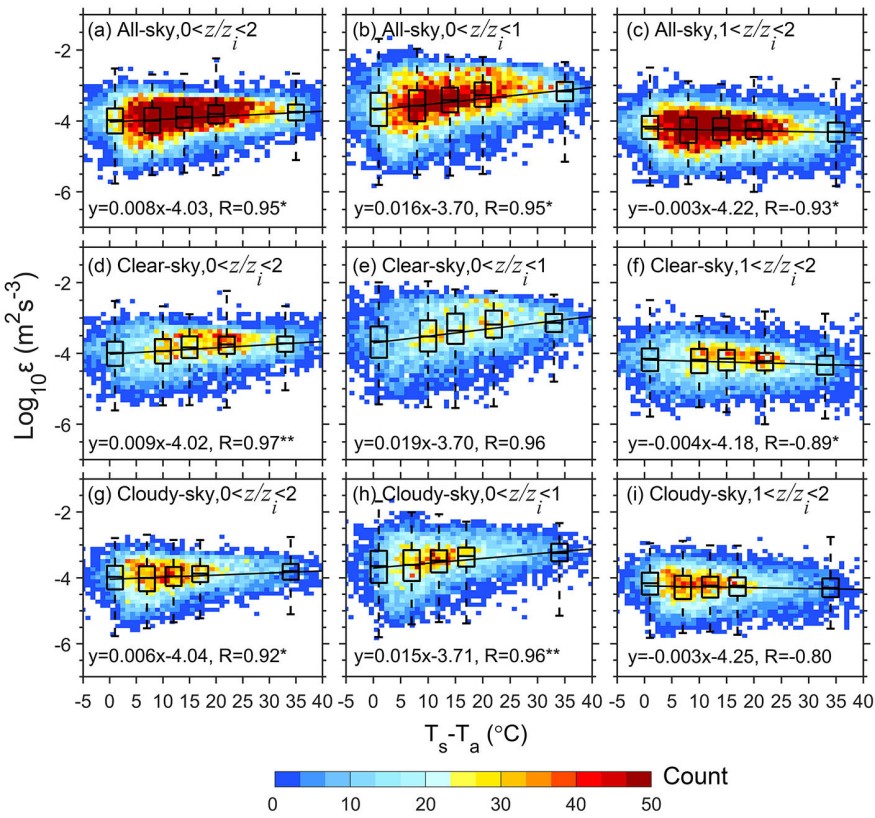

**Figure 6.** Scatter plots (blue dots) of $Log_{10}\varepsilon$ estimated from the measurements of RWPs
in the whole lower troposphere ($0.2 < z/z_i < 2.0$, a, d, g), in the PBL ($0.2 < z/z_i < 1.0$, b, e, h)
and above the PBL ($1.0 < z/z_i < 2.0$, c, f, i) over the TP as a function of $T_s - T_a$ under all-
sky (a-c), clear-sky (d-f) and cloudy-sky conditions (g-i), respectively. Also overlaid are
their corresponding box and whisker plots and regression linear equations and correlation
coefficients in each panel, where all $T_s - T_a$ samples are divided into five bins, each of
which has the same sample size. Note that the median is shown as a line whereas the outer
boundaries of the boxes represent 25 and 75 quartiles and the dashed lines present
interquartile range (IQR). The superscripts * and ** for R indicate that the regression slopes
are statistically significant at $p < 0.05$ and $p < 0.01$ levels, respectively.





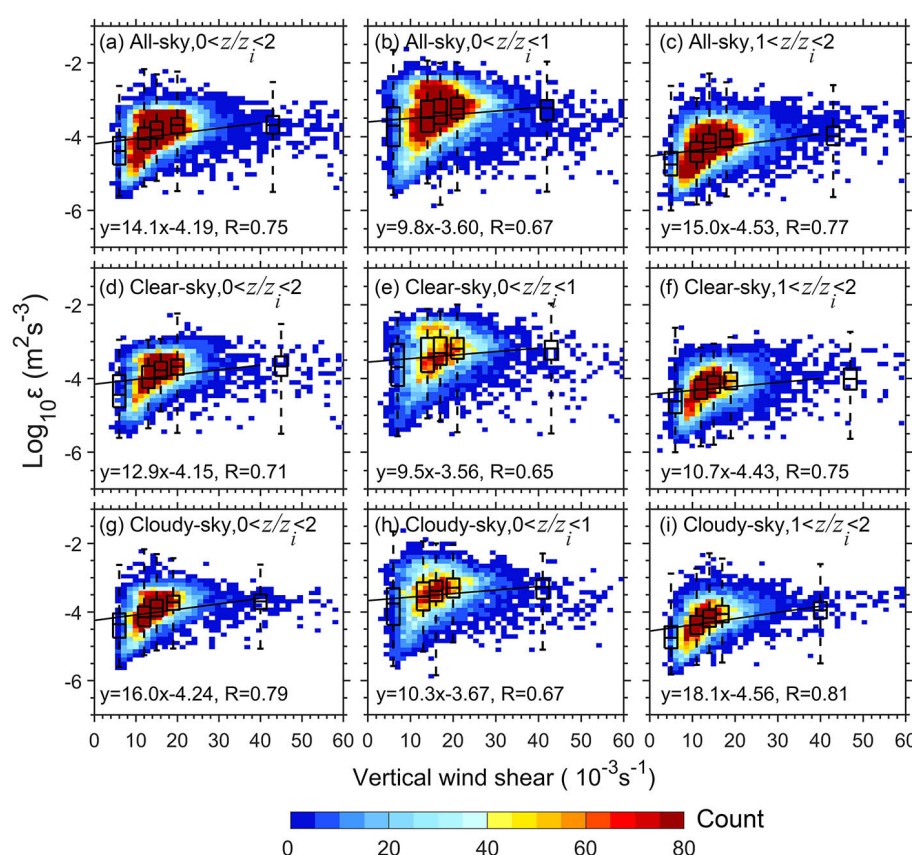

**Figure 7.** Same as Figure 5, but for $Log_{10}\varepsilon$ as a function of vertical wind shear (VWS).



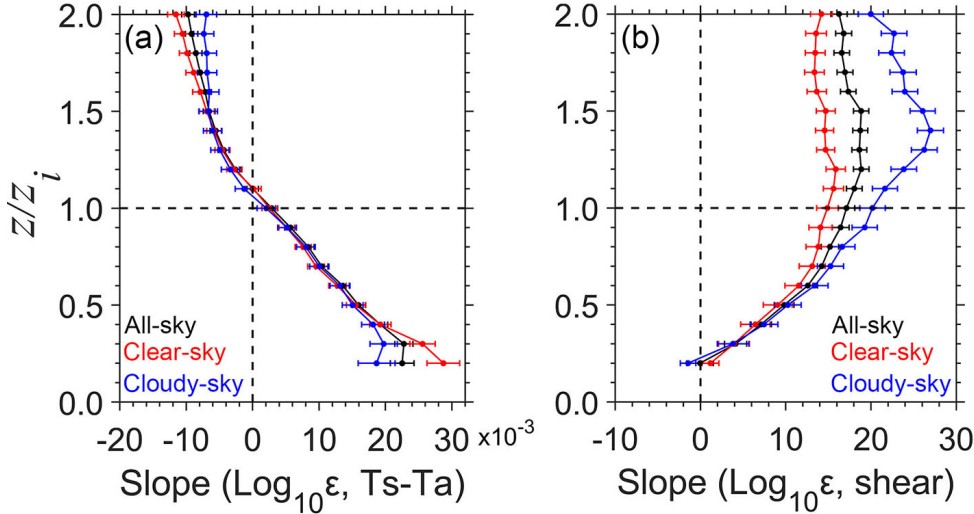


**Figure 8.** The vertical profiles of least squares regression slope between $Log_{10}\varepsilon$ and

$T_s - T_a$ (a) and vertical wind shear (b) over the TP under all-sky (black), clear-sky (red)
and cloudy-sky (blue) conditions, respectively.

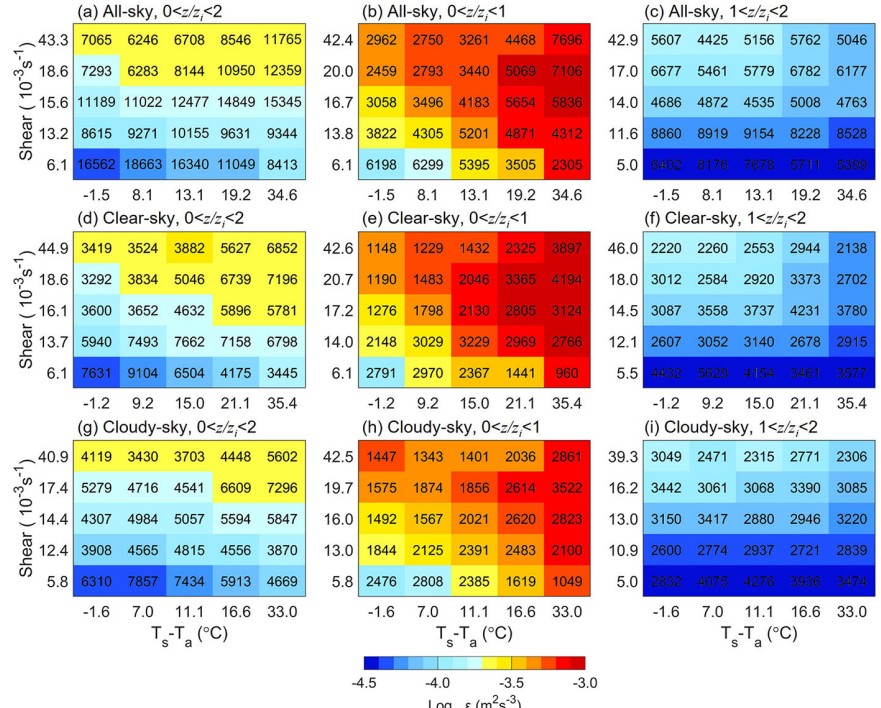


**Figure 9.** Joint dependence of $Log_{10}\varepsilon$ (color shading) on the vertical wind shear and $T_s - T_a$ within and above the PBL (a, d, g), in the PBL (b, e, h) and above the PBL (c, f, i) over the TP under all-sky (a-c), clear-sky (d-f) and cloudy-sky (g-i) conditions, respectively. The number given in each panel is the total number of samples used.



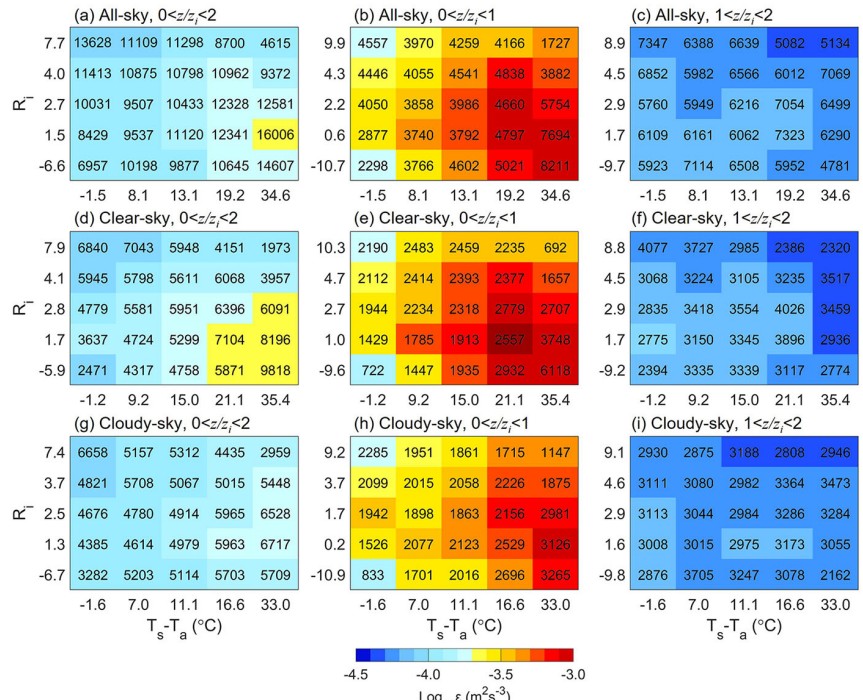

**Figure 10.** Joint dependence of $Log_{10}\varepsilon$ (color shading) on the gradient Richardson number ($Ri$) and $T_s - T_a$ in and above the PBL (a, d, g), in the PBL (b, e, h) and above the PBL (c, f, i) over the TP under all-sky (a-c), clear-sky (d-f) and cloudy-sky (g-i) conditions, respectively. The number given in each panel is the total number of samples used.