# Peer review of "Elucidating the boundary layer turbulence dissipation rate using high-resolution measurements from a radar wind profiler network over the Tibetan Plateau Deli Meng1,2, Jianping Guo1,3\* Xiaoran Guo1\*, Yinjun Wang1, Ning Li1, Yuping Sun1</s"

_EGUsphere, 2024_

## Referee Comment (RC2)

Review of "Elucidating the boundary layer turbulence dissipation rate using high-resolution measurements from a radar wind profiler network over the Tibetan Plateau" by Meng et al, submitted for publication in ACP

April 2024

The authors use one year of data from the Chinese radar wind profiler (RWP) network at six sites at the Tibetan Plateau (TP) to evaluate the evolution of turbulence intensity (turbulence dissipation rate $\varepsilon$) and planetary boundary layer (PBL) height (zi) throughout the day. First they show the daily variation of these parameters, averaged over the whole year for each daytime hour, at each of these six sites and point out the large differences between the two site in the North of the TP (with high turbulence intensity) and those in the south and east of the TP (low turbulence intensity). The differences are ascribed to different land cover, but no detail is presented. Next the data from all six stations are averaged per hour and effects of surface-air temperature (deltaT), vertical wind shear (VWS) and cloud cover are discussed. The authors conclude that deltaT has the largest influence in the PBL and VWS has the largest influence above the PBL.

The paper provides information on the spatial and temporal variation of turbulence and PBL and effects of some factors across the TP and is suitable for publication in ACP, subject to revisions addressing the general and detailed comments below, including suggestions for technical corrections.

**General comments**. In view of the differences between the six sites, in particular as regards turbulence dissipation rate (Fig. 2), what is the justification to average the data over all six sites? Would it not be more reasonable to analyze the data for each site and compare the results? This could provide information on other factors influencing the turbulence characteristics. Along the same lines, cloud cover and especially surface and air temperature vary strongly with season. Why, instead of discussing seasonal variation, are the data averaged over the whole year? Likely, the seasonal variations also vary between the six sites and the data for each site might provide constraints depending on local conditions which now are hidden in the large amount of data but for different conditions. Furthermore, the effect of cloudy vs clear sky likely affects the deltaT (and the surface and air temperature) and thus would provide more information than looking at all data together. This likely also explains the relatively small difference between cloudy and clear sky zi of only 117 m (line 310).

Another comment is the conclusion the "incapability of analyzing the effect of wind shear on $\varepsilon$ below 0.5 km AGL in the following section" (line 376). With a maximum zi of about 2 km (figs. 2 & 4) this implies that z/zi needs to be at least 0.5/2=0.25, and preferably should be a function of zi. However z/zi is sometimes >0, sometimes >0.2 (For instance Fig. 6 and caption and text use different measures, but it is throughout the whole MS) and lsq fit seem to be made over different ranges (Fig 6 and Table 2).

The authors define AGL as "above sea level" (line 140) but use AGL also when they mean above ground level. I suggest to define above sea level as ASL and above ground level as AGL and check the paper when each is meant to be used.

The authors use "trend" but do not derive any trend. The difference between two data points (line 251) cannot be called a trend, in particular when these data are taken about 1500 km apart and nothing is known about the variation in between , may be just say that in the east the one-year averaged turbulence is smaller than at the western site? Note that "trend" is also used wrongly at other instances to indicate an increase or decrease.

**Detailed comments:**

spatial discrepancy > do you mean these a large difference between the six sites?

difference in land do clouds suppress turbulence? Or does solar irradiation heat the surface which creates a larger deltaT and thus turbulence (as is discussed in the paper, see also line 47).

change to: impact on the forecast skill of weather and climate models

70/71    hard for radiosondes and ultrasonic anemometers …. of atmospheric ..

elevation larger than 4000 m change could to can change bubbling to thermals of warm air change to : understanding the …

change influential to influencing change to: flux promotes the …

change to: clouds tend to suppress change to: China using fine …

change to: compared to clear …

change to: PBL contributors to ..

change to: in turn influences ..

change "elusive" to "unclear"

117/118        change to: Coincidently, the RWP network in China provides us a valuable …

of the RWP

and detailed information

ASL (see general comments)

Dunhuang?

signal to noise is this a hypothesis or are these assumptions?

excluding the above from the turbulent between the Earth's surface and the atmosphere above affecting cloud profile is greater presents

TP ranges from meridional or latitudinal?

reaches values up to replace "least magnitude " with "lowest value"

Par. Starting at line 272: is there an explanation why zi at Dingri is so much higher than at any of the other 5 sites?

replace "both" with "the"

vegetated terrain at the Ganzi the sentence suggests that Fig.1 shows vegetation, but the locations seems to be overlaid on an elevation map in the PBL

the sentence starting with "Thus, " : some word seems to be missing (spatial variation?) but the sentence does not make sense: is the dissipation relevant to the surface type or does the surface type influence the turbulence?

PBL properties?

299-304 it is not mentioned to which sky conditions this applies

$\varepsilon$ ranges from

350-351 does Table 2 show scatter plots or …. please correct and explain what table 2 shows, and also why the lower limit of z/zi is 0.2 whereas in the plots you use 0. Obviously the range influences the lsq fits, as the comparison between the eqs in Table 2 and Fig. 6 shows. However, Fig 8 shows that the lowest data point is for z/zi =0.2. Please discuss this in the text, and if no data exist below z/zi=0.2, all figures and text mentioning z/zi>0 needs to be corrected.

-364 The data and discussion clearly show the effect of clouds on the turbulence in the PBL. However, the question arises whether clouds reduce the solar irradiation at the surface and thus surface heating and thus deltaT. The extent to which deltaT changes depends on COT and cloud cover. Therefore I would suggest that deltaT is the governing parameter rather than cloud cover.

As mentioned in the text (see also general comments), VWS influences turbulence within the PBL, but it can be determined only in the upper part (>500 m). Hence the summary sentence on lines 383-384 should more carefully formulated to do justice to the detail presented in the above.

Also in the rest of the text, the conclusions on the effect of VWS within the PBL need to be more carefully formulated (see also general comment).

decreases with height (remove trend)

linear variation of the slope from the lower PBL to the top of the PBL. Within the PBL< the slope is positive, above the PBL ...

411-413 Fig 8 clearly shows the influence of cloud cover on the deltaT and the effect of the surface heating on the turbulence in the lower half of the PBL (z/zi<0.5, while higher in the PBL the surface effect has dampened and there is no difference between clear and cloudy sky.

the slope decreases with height buoyant and mechanic forcing at the Minfeng

472-473 similar to comment on line 282

---

## Author Comment (AC1)

**Responses to the reviewer #2' comments**

**Reviewer #2:**

The authors use one year of data from the Chinese radar wind profiler (RWP) network at six sites at the Tibetan Plateau (TP) to evaluate the evolution of turbulence intensity (turbulence dissipation rate $\varepsilon$) and planetary boundary layer (PBL) height (zi) throughout the day. First, they show the daily variation of these parameters, averaged over the whole year for each daytime hour, at each of these six sites and point out the large differences between the two sites in the North of the TP (with high turbulence intensity) and those in the south and east of the TP (low turbulence intensity). The differences are ascribed to different land cover, but no detail is presented. Next the data from all six stations are averaged per hour and effects of surface-air temperature (deltaT), vertical wind shear (VWS) and cloud cover are discussed. The authors conclude that deltaT has the largest influence in the PBL and VWS has the largest influence above the PBL.

*Reply: We appreciate the thoughtful and excellent comments made by the reviewer. We have tried as much as possible to address all the concerns and have revised the manuscript accordingly. The reviewer' comments are written in normal font, and our point-to-point responses to the editor' comments are in blue italics.*

**General comments.**

1. In view of the differences between the six sites, in particular as regards turbulence dissipation rate (Fig. 2), what is the justification to average the data over all six sites? Would it not be more reasonable to analyze the data for each site and compare the results? This could provide information on other factors influencing the turbulence characteristics. Along the same lines, cloud cover and especially surface and air temperature vary strongly with season. Why, instead of discussing seasonal variation, are the data averaged over the whole year? Likely, the seasonal variations also vary between the six sites and the data for each site might provide constraints depending on local conditions which now are hidden in the large amount of data but for different conditions. Furthermore, the effect of cloudy vs clear sky likely affects the deltaT (and the surface and air temperature) and thus would provide more information than looking at all data together. This likely also explains the relatively small difference between cloudy and clear sky zi of only 117 m (line 310).

*Response: Per your kind suggestions, we firstly added a discussion on the spatial and temporal distribution characteristics of turbulent dissipation rate ($\varepsilon$) at six RWP stations over the Tibetan Plateau (Fig. S1) in section 3.1 of this revised manuscript. Then, the relationships between the subsurface, surface-air temperature difference ($T_s - T_a$), vertical wind shear (VWS) and $\varepsilon$ are examined separately for each RWP station at the heights ranging from 0.5 to 3.0 km, which is shown in Fig. 2, Fig. S2, and Fig. S3, respectively.*

*It has been found that there exists diurnal $\varepsilon$ variation in planetary boundary layer (PBL) at the six RWP stations over the TP in section 3.1. To better reveal the mechanism how a myriad of geophysical parameters affect turbulence under clear- and cloudy-sky conditions, the height-revolved $\varepsilon$ retrievals are further normalized by the average PBL height from section 3.2. The effect of clouds on the vertical structure of turbulence at different RWP stations is given in Fig. S4, and the effects of $T_s - T_a$ and VWS on $\varepsilon$ (Fig. S5) were further studied.*

*On the seasonal scale, the turbulence at the six RWP stations is characterized by significant variability, which is shown in Fig. S1. To be more specific, $\varepsilon$ reaches the maximum in summer with the highest $z_i$, while touches the minimum in winter at Minfeng and Jiuquan. At the remaining four stations, the strongest $\varepsilon$ is found in spring, as opposed to the weakest $\varepsilon$ in autumn.*

[Figure]

*Figure S1. Spatial distribution of the seasonal evolution of the vertical profile of logarithmic turbulence dissipation rate ($Log_{10}\varepsilon$ in color shading, unit: $m^2\ s^{-3}$) at 120 m vertical resolution and 6 min intervals, and hourly mean planetary boundary layer height ($z_i$, black line, unit: km) during daytime under all-sky conditions from 0900 to 1700 LST for the period September 2022 to October 2023 as retrieved from the profiling measurements at six RWP stations over the TP. The vertical bars indicate the 0.5 standard deviations for $z_i$.*

*As shown in Fig S2, it can be seen that there is a positive correlation between the $T_s - T_a$ and $\varepsilon$ at heights from 0.5 to 3.0 km, indicating that the thermal effect of the $T_s - T_a$ can promote the development of turbulence under all-sky conditions. However, the relationship varies significantly between each RWP station. The slope values of the regression coefficients for the other five RWP stations, except for Hongyuan are all greater than 0.015. The maximum slope values are observed at Lijiang (0.029) and*

*Dingri (0.027) in the southern TP, as compared with the minimum slope of 0.007 at Hongyuan. This suggests that near-surface thermal properties have nothing to do with $\varepsilon$ at Hongyuan in the eastern TP.*

[Figure]

*Figure S2. Scatter plots of $Log_{10}\varepsilon$ at heights from 0.5 to 3.0 km as a function of $T_s - T_a$ at six RWP stations over the TP during daytime under all-sky conditions from 0900 to 1700 LST for the period September 2022 to October 2023. The superscript ** for R indicates that the regression slope is statistically significant at $p < 0.01$ level.*

*The VWS is also found to positively correlate with $\varepsilon$ at heights from 0.5 to 3.0 km under all-sky conditions (Fig. S3), indicating that the dynamic effect of VWS would promote the development of turbulence. However, the relationship between VWS and $\varepsilon$ varies significantly among different RWP stations. In terms of the slope value, it can reach 79.34 at Lijiang, followed by Hongyuan (68.56) and Dingri (61.82), and only 6.10 and 10.29 at Minfeng and Jiuquan., Although the slope value at Ganzi is only 9.5,*

*it can be up to 41.64 on the interval of 0-0.04s⁻¹ for the VWS. It can be seen that the dynamics of VWS significantly influences the turbulence and is significantly stronger in the southern and eastern TP than in the northern TP.*

*Figure S4 further shows the distinct spatial variability of cloud effect on ε across the six RWP stations. Particularly, the turbulence is weakened by clouds within the PBL at Minfeng and Jiuquan in the northern TP, as opposed to the enhanced ε within the PBL at Ganzi and Lijiang. This suggests that the cloud impact on ε is much complicated than expected. One of the reasons could be concerned with the cloud life stage, which is not dealt with in this present study. On top of the life stage, the cloud impact on ε, in combination with $T_s - T_a$ and VWS, exhibits a distinct altitude dependence, differing by RWP stations (Fig. S5).*

*The above-mentioned points have been incorporated in this revised manuscript.*

[Figure]

*Figure S3. Scatter plots of $Log_{10}\varepsilon$ at heights from 0.5 to 3.0 km as a function of vertical wind shear (VWS) at six RWP stations over the TP during daytime under all-sky conditions from 0900 to 1700 LST for the period September 2022 to October 2023. The superscript \*\* for R indicates that the regression slope is statistically significant at $p <$ 0.01 level.*

[Figure]

*Figure S4. Normalized contoured frequency by altitude diagram (NCFAD) for the difference of $\varepsilon$ between cloudy-sky and clear-sky conditions ($\Delta\varepsilon$) at six RWP stations over the TP from 0900 to 1700 LST for the period September 2022 to October 2023. Note that $z_i$ denotes the depth of the PBL, the height (z) and turbulence dissipation rate ($\varepsilon$) is normalized by $z_i$ in order to give a nondimensional vertical coordinate in the form of $z/z_i$.*

[Figure]

*Figure S5. The vertical profiles of least squares regression slope between $Log_{10}\varepsilon$ and $T_s - T_a$ and VWS under all-sky (black), clear-sky (red) and cloudy-sky (blue) conditions at six RWP stations over the TP from 0900 to 1700 LST for the period September 2022 to October 2023. Note that $z_i$ denotes the depth of the PBL, the height ($z$) and turbulence dissipation rate ($\varepsilon$) is normalized by $z_i$ in order to give a nondimensional vertical coordinate in the form of $z/z_i$.*

2. Another comment is the conclusion the "incapability of analyzing the effect of wind shear on $\varepsilon$ below 0.5 km AGL in the following section" (line 376). With a maximum zi of about 2 km (figs. 2 & 4) this implies that z/zi needs to be at least 0.5/2=0.25, and preferably should be a function of zi. However, z/zi is sometimes >0, sometimes >0.2 (For instance Fig. 6 and caption and text use different measures, but it is throughout the whole MS) and lsq fit seem to be made over different ranges (Fig 6 and Table 2).

*Response: Good point! In this revision, the starting height of* z/zi *has been changed to 0.3, given the minimum value of 0.25. For more details, please refer to Figs.5-10.*

3. The authors define AGL as "above sea level" (line 140) but use AGL also when they mean above ground level. I suggest to define above sea level as ASL and above ground level as AGL and check the paper when each is meant to be used.

*Response: Above sea level is a typo, and it has been corrected to Above ground level in this revision. Thanks.*

4. The authors use "trend" but do not derive any trend. The difference between two data points (line 251) cannot be called a trend, in particular when these data are taken about 1500 km apart and nothing is known about the variation in between, may be just say that in the east the one-year averaged turbulence is smaller than at the western site? Note that "trend" is also used wrongly at other instances to indicate an increase or decrease.

*Response: Per your suggestion, it has been revised to "It is apparent that $\varepsilon$ exhibits a larger west-east and north-southern spatial discrepancy under all-sky conditions. In terms of the latitudinal variation, the one-year averaged $\varepsilon$ at the RWP stations in the east part of TP is smaller than in the western part of TP."*

*Besides, to clarify the characteristics of the vertical profiles of least squares regression slope values between $Log_{10}\varepsilon$ and $T_s - T_a$ and VWS, we have modified the corresponding descriptions in this revision.*

**Detailed comments:**

spatial discrepancy > do you mean these a large difference between the six stations?

*Response: Yes. It has been clarified as "..exhibits a large spatial discrepancy over the six RWP stations over the TP"*

difference in land

*Response: Revised as suggested.*

do clouds suppress turbulence? Or does solar irradiation heat the surface which creates a larger deltaT and thus turbulence (as is discussed in the paper, see also line 47).

*Response: Per your suggestion, it has been revised to "This could be the cooling effect by cloud that reduces the solar irradiation reaching the surface."*

change to: impact on the forecast skill of weather and climate models

*Response: Revised as suggested.*

70/71 hard for radiosondes and ultrasonic anemometers …. of atmospheric .

*Response: Revised as suggested.*

elevation larger than 4000 m

*Response: It has been revised to "greater than 4000 m".*

change could to can

*Response: Revised as suggested.*

change bubbling to thermals of warm air

*Response: Revised as suggested.*

change to : understanding the …

*Response: revised as suggested.*

change influential to influencing

*Response: Revised as suggested.*

change to: flux promotes the …

*Response: Revised as suggested.*

change to: clouds tend to suppress

*Response: Revised as suggested.*

change to: China using fine …

*Response: Revised as suggested.*

change to: compared to clear …

*Response: revised as suggested.*

change to: PBL contributors to .

*Response: Revised as suggested.*

change to: in turn influences ..

*Response: revised as suggested.*

change "elusive" to "unclear"

*Response: Revised as suggested.*

117/118 change to: Coincidently, the RWP network in China provides us a valuable …

*Response: Revised as suggested.*

of the RWP

*Response: Revised as suggested.*

and detailed information

*Response: Revised as suggested.*

ASL (see general comments)

*Response:               Revised               as               suggested.*

Dunhuang?

*Response: "Dunhuang" has been corrected to "Jiuquan".*

signal to noise

*Response: Revised as suggested.*

is this a hypothesis or are these assumptions?

*Response: It should be assumptions, which has been clarified in this revision.*

excluding the above

*Response: Revised as suggested.*

from the turbulent

*Response: Revised as suggested.*

between the Earth's surface and the atmosphere above

*Response: Revised as suggested.*

affecting cloud

*Response: Revised as suggested.*

profile is greater

*Response: Revised as suggested.*

presents

*Response: Revised as suggested.*

TP ranges from

*Response: Revised as suggested.*

meridional or latitudinal?

*Response: This should be "meridional", and we have revised it as suggested.*

reaches values up to

*Response: Revised as suggested.*

replace "least magnitude " with "lowest value"

*Response: Revised as suggested.*

Par. Starting at line 272: is there an explanation why zi at Dingri is so much higher than at any of the other 5 sites?

*Response: This point has been added in revised manuscript:*

*"Of the six RWP stations, Dingri is located in the northern foothills of the Himalayas with an altitude of over 4300 m, where the bare land type results in a large surface sensible heat flux. This, together with the lowest atmospheric density, leads to the highest daytime mean value of $z_i$ up to 2.10 km (Wang et al., 2015)."*

replace "both" with "the"

*Response: Revised as suggested.*

vegetated terrain at the Ganzi

*Response: Revised as suggested.*

the sentence suggests that Fig.1 shows vegetation, but the locations seems to be overlaid on an elevation map

*Response: Revised in the figure caption, per your suggestion.*

in the PBL

*Response: Revised as suggested.*

the sentence starting with "Thus, " : some word seems to be missing (spatial variation?) but the sentence does not make sense: is the dissipation relevant to the surface type or does the surface type influence the turbulence?

*Response: You are right. It has been rephrased to "Therefore, we argue that the spatial and temporal variation of daytime ε over the TP are affected by the underlying surface type and air density."*

PBL properties?

*Response: Yes, and it has been revised as suggested.*

299-304 it is not mentioned to which sky conditions this applies

*Response: It has been clarified as "under cloud-sky conditions".*

ε ranges from

*Response: The grammar error has been corrected.*

350-351 does Table 2 show scatter plots or …. please correct and explain what table 2 shows, and also why the lower limit of z/zi is 0.2 whereas in the plots you use 0. Obviously the range influences the lsq fits, as the comparison between the eqs in Table 2 and Fig. 6 shows. However, Fig 8 shows that the lowest data point is for z/zi =0.2. Please discuss this in the text, and if no data exist below z/zi=0.2, all figures and text mentioning z/zi>0 needs to be corrected

*Response: The range of z/zi is revised to 0.3 to 2.0 throughout the revised manuscript. For more details, please see our response to General comment 2,*

-364 The data and discussion clearly show the effect of clouds on the turbulence in the PBL. However, the question arises whether clouds reduce the solar irradiation at the surface and thus surface heating and thus deltaT. The extent to which deltaT changes depends on COT and cloud cover. Therefore, I would suggest that deltaT is the governing parameter rather than cloud cover.

*Response: Good point! Yes, you are right. We have revised in the revised manuscript.*

As mentioned in the text (see also general comments), VWS influences turbulence within the PBL, but it can be determined only in the upper part (>500 m). Hence the summary sentence on lines 383-384 should more carefully formulated to do justice to the detail presented in the above.

*Response: Thanks for your careful checks, we have modified the corresponding descriptions in this revised manuscript per your kind suggestions.*

Also in the rest of the text, the conclusions on the effect of VWS within the PBL need to be more carefully formulated (see also general comment).

*Response: Per your kind suggestion, we have revised the rest of the paper in the revised manuscript.*

decreases with height (remove trend)

*Response: Revised as suggested.*

linear variation of the slope from the lower PBL to the top of the PBL. Within the PBL< the slope is positive, above the PBL ...

*Response: Revised as suggested.*

411-413 Fig 8 clearly shows the influence of cloud cover on the deltaT and the effect of the surface heating on the turbulence in the lower half of the PBL (z/zi

*Response: revised as suggested.*

the slope decreases with height

*Response: Revised as suggested.*

buoyant and mechanic forcing

*Response: "buoyant and mechanistic" has been revised to "buoyant and mechanic forcing".*

at the Minfeng

*Response: Revised as suggested.*

472-473 similar to comment on line 282

*Response: Revised as suggested.*

PBL in clear-sky

*Response: It has been revised to "in the PBL under clear-sky conditions".*

remains known or unknown?

*Response: It has been revised to "unknown"?.*

---

## Author Comment (AC2)

**Responses to the reviewer #1' comments**

**Reviewer #1:** The manuscript aims to advance our understanding of the planetary boundary layer (PBL) turbulence and evolution over the Tibetan Plateau (TP) by utilizing high-resolution radar wind profiler (RWP) data. The study demonstrates the spatiotemporal variations and underlying mechanisms of turbulence dissipation rates in the PBL. The authors also provide detailed analyses of how land cover, radiation, and vertical wind shear influence PBL turbulence. Overall, this manuscript is well organized with significant scientific advancement. I recommend the publication of this paper in Atmospheric Chemistry and Physics subject to minor revisions.

*Response: We are glad to receive your positive and encouraging comments, which are invaluable in improving the quality of our manuscript. For clarity purpose, here we have listed the reviewer's comments in black plain font, followed by our response in blue italics.*

**Minor comments:**

1. In the methodology section, the authors may include a discussion of the potential uncertainties and limitations of turbulence dissipation rate or boundary layer height from the RWP. It will help readers have a better understanding of the strengths and limitations of RWP measurements.

*Response: Good point. Per your suggestion, the potential uncertainties and limitations of turbulence dissipation rate (ε) and boundary layer height from the RWP have been discussed in this revision.*

*(1) The uncertainties and limitations of turbulence dissipation rate (ε) from the RWP is discussed in section 2.3.1, which is shown as follows.*

*"One caveat of the above-mentioned methods used to estimate ε lies in its sensitivity to the uncertainty in measuring horizontal wind speed, and the occurrence of negative value of $\sigma_t^2$ (resulting in negative ε and invalid retrieval), which is previously documented (e.g., Chen et al., 2021; McCaffrey et al., 2017). Also noteworthy is that ε estimates derived from the RWP lacks validation against in situ ε measurements from*

*sonic anemometer in the aircraft or tower. This is another factor causing uncertainties that needs to be addressed in the future."*

*(2) The uncertainties and limitations of PBL height from the RWP is discussed in section 2.3.2, which is shown as follows.*

*"It is not optimal to retrieve $z_i$ directly from the RWP measurements during nighttime, when the weaker turbulence and greater SNR tend to result in an overestimation of $z_i$ (Duncan et al., 2022). Therefore, the $z_i$ estimation using the ITM algorithm is merely applicable in the daytime convective PBL (Bianco et al., 2008; Collaud Coen et al., 2014). Besides, the presence of clouds is proved to bring about uncertainty in $z_i$ retrievals from the ITM, due to the challenge in identifying the peak from the NSNR profile (Angel et al., 2024). "*

***References:***

*Chen, Z., Tian, Y., Wang, Y., Bi, Y., Wu, X., Huo, J., Pan, L., Wang, Y., and Lü, D.: Turbulence parameters measured by the Beijing mesosphere–stratosphere–troposphere radar in the troposphere and lower stratosphere with three models: comparison and analyses, Atmos. Meas. Tech., 15, 4785–4800, https://doi.org/10.5194/amt-15-4785-2022, 2022.*

*Collaud Coen, M., Praz, C., Haefele, A., Ruffieux, D., Kaufmann, P., and Calpini, B.: Determination and climatology of the planetary boundary layer height above the Swiss plateau by in situ and remote sensing measurements as well as by the COSMO-2 model, Atmos. Chem. Phys., 14, 13205–13221, https://doi.org/10.5194/acp-14-13205-2014, 2014.*

*Duncan Jr., J. B., Bianco, L., Adler, B., Bell, T., Djalalova, I. V., Riihimaki, L., Sedlar, J., Smith, E. N., Turner, D. D., Wagner, T. J., and Wilczak, J. M.: Evaluating convective planetary boundary layer height estimations resolved by both active and passive remote sensing instruments during the CHEESEHEAD19 field campaign, Atmos. Meas. Tech., 15, 2479–2502, https://doi.org/10.5194/amt-15-2479-2022, 2022.*

*McCaffrey, K., Bianco, L., and Wilczak, J. M.: Improved observations of turbulence dissipation rates from wind profiling radars, Atmos. Meas. Tech., 10, 2595-2611,*

*https://doi.org/10.5194/amt-10-2595-2017, 2017.*

*Angel A C, Manoj M G. A novel method of estimating atmospheric boundary layer height using a 205 MHz VHF radar. Sci. Total. Environ., https://doi.org/10.1016/j.scitotenv.2023.168109, 907: 168109, 2024.*

2. The introduction mentions that "Also, cloud radiative effects are found to be another significant factor to modulate the evolution of daytime PBL turbulence (Bodenschatz et., 2010)." However, the references provided for this issue are insufficient. I suggest the authors include more and acknowledge the previous work on this issue, such as the impacts of cloud radiative forcing on the morning transition from a stable to an unstable boundary layer.

*Response: Per your kind suggestion, more previous work on this topic have been cited in this revision, which is shown as follows.*

*"Except for the above-mentioned thermal and dynamic effects, cloud radiative effect is found to be another significant factor that can dramatically modulate the evolution of daytime PBL turbulence (Bodenschatz et., 2010; Davis et al., 2020). For instance, cloud radiative forcing accounts for the rapid morning transition from stable to unstable PBL, thereby notably affecting the diurnal variation of the PBL (Su et al., 2023). Notably, longwave radiative cooling at the top of stratocumulus clouds can enhance turbulent diffusion within the stratocumulus topped PBL (Sun et al., 2016). A recent observational study suggests that cloud radiative cooling contributed about 32% to turbulent mixing even near the surface (Huang et al., 2020). In other words, cloud radiative processes, including entrainment and radiative cooling, can affect the TKE in the atmosphere (Nicholls et al., 1986; Sedlar et al., 2022; Chechin et al., 2023)."*

*References:*

*Chechin, D. G., Lüpkes, C., Hartmann, J., Ehrlich, A., and Wendisch, M.: Turbulent structure of the Arctic boundary layer in early summer driven by stability, wind shear and cloud-top radiative cooling: ACLOUD airborne observations, Atmos. Chem. Phys., 23, 4685-4707, https://doi.org/10.5194/acp-23-4685-2023, 2023.*

*Davis E V, Rajeev K, and Mishra M K: Effect of clouds on the diurnal evolution of the*

*atmospheric boundary-layer height over a tropical coastal station. Bound.-Layer Meteor., 175: 135-152, https://doi.org/10.1007/s10546-019-00497-6, 2020.*

*Huang, T., Yim, S. H. L., Yang, Y., Lee, O. S. M., Lam, D. H. Y., Cheng, J. C. H., and Guo, J.: Observation of turbulent mixing characteristics in the typical daytime cloud-topped boundary layer over Hong Kong in 2019, Remote Sens., 12, 1533, https://doi.org/10.3390/RS12091533, 2020.*

*Nicholls, S.: The dynamics of stratocumulus: Aircraft observations and comparisons with a mixed layer model, Q. J. Roy. Meteor. Soc., 110, 783–820, https://doi.org/10.1002/qj.49711046603, 1984.*

*Su, T. N., Li, Z. Q., and Zheng, Y. T.: Cloud-Surface Coupling Alters the Morning Transition From Stable to Unstable Boundary Layer, Geophys. Res. Lett., 50, 9, https://doi.org/10.1029/2022gl102256, 2023.*

*Sun, W., Li, L., and Wang, B.: Reducing the biases in shortwave cloud radiative forcing in tropical and subtropical regions from the perspective of boundary layer processes, Sci. China Earth Sci., 59, 1427–1439, https://doi.org/10.1007/s11430-016-5290-z, 2016.*

3. The authors use PBL turbulence as the key concept. I suggest the authors mention that the scope of PBL turbulence beyond the current discussions, such as heat fluxes, vertical velocity, entrainment, etc.

*Response: Good point! Per your suggestion, we have expanded on the scope of PBL turbulence in section 4, which is shown as follows:*

*"On top of this, the role of roughness length, vertical velocity, and entrainment remains unknown in the variation and evolution of atmospheric turbulence, which warrants further in-depth studies based on intensive field campaigns, in combination with theoretical analysis and numerical simulation experiments in the future."*

Line 33: "large spatial discrepancy" -> "a large spatial discrepancy"

*Response: Corrected as suggested.*

Line 36: "the difference of" -> "the difference in"

*Response: Corrected as suggested.*

Line 62: "have great impact" -> "have great impacts"

*Response: Corrected as suggested.*

Line 69: "the RWP exhibit" -> "the RWP exhibits"

*Response: Corrected as suggested.*

Line 485: "The slope values of e against VWS is" -> "The slope values of e against VWS are""

*Response: Corrected as suggested.*